# Performance Improvement of Machine Learning Model Using Autoencoder to Predict Demolition Waste Generation Rate

**Gi-Wook Cha** [1] , **Won-Hwa Hong** [2] **and Young-Chan Kim** [3],*

1  School of Science and Technology Acceleration Engineering, Kyungpook National University, Daegu 41566, Republic of Korea
2  School of Architectural, Civil, Environmental and Energy Engineering, Kyungpook National University, Daegu 41566, Republic of Korea
3  Division of Smart Safety Engineering, Dongguk University Wise Campus, 123 Dongdae-ro, Gyeongju 38066, Republic of Korea
*  Correspondence: yyoungchani@gmail.com

**Abstract:** Owing to the rapid increase in construction and demolition (C&D) waste, the information of waste generation (WG) has been advantageously utilized as a strategy for C&D waste management. Recently, artificial intelligence (AI) has been strategically employed to obtain accurate WG information. Thus, this study aimed to manage demolition waste (DW) by combining three algorithms: artificial neural network (multilayer perceptron) (ANN-MLP), support vector regression (SVR), and random forest (RF) with an autoencoder (AE) to develop and test hybrid machine learning (ML) models. As a result of this study, AE technology significantly improved the performance of the ANN model. Especially, the performance of AE (25 features)–ANN model was superior to that of other non-hybrid and hybrid models. Compared to the non-hybrid ANN model, the performance of AE (25 features)–ANN model improved by 49%, 27%, 49%, and 22% in terms of the MAE, RMSE, $R^2$, and R, respectively. The hybrid model using ANN and AE proposed in this study showed useful results to improve the performance of the DWGR ML model. Therefore, this method is considered a novel and advantageous approach for developing a DWGR ML model. Furthermore, it can be used to develop AI models for improving performance in various fields.

**Keywords:** artificial intelligence; autoencoder; demolition waste; hybrid model; machine learning; waste management

## 1. Introduction

The World Bank (2018) predicted that the annual volume of municipal solid waste (MSW) generated in cities will increase to 3.4 billion tons in 2050 [1]. In addition to the deterioration of the urban environment, the increase in waste poses various environmental and health risks, such as groundwater pollution, land degradation, increased cancer incidence, child mortality, and birth defects [2]. Therefore, governments worldwide and researchers in related industries have been striving to realize effective waste management (WM) strategies, and in recent years, advanced technologies and intelligent systems have been introduced in an attempt to manage waste.

Accounting for 35–40% of global waste, construction and demolition waste (C&DW) is increasing at an alarming rate and has emerged as a concern for the national economy and sustainable development goals [3–5]. In addition, the generation of C&DW is steadily increasing [6–8], wherein demolition waste (DW) accounts for 70–90% of C&DW [9,10]. As such, copious amounts of generated C&DW create considerable problems and increase the social and environmental burden through their deleterious effects on the environment. Therefore, pertinent management of C&DW is essential, and as a potential WM strategy, several researchers have utilized the information on waste generation (WG) as an advantageous tool for C&DW management. However, WM processes involve complex systems

influenced by several factors, and obtaining adequate and satisfactory information for WM is a challenging task.

Consequently, the application of artificial intelligence (AI) technology has been actively considered in recent years to derive useful and satisfactory information for WM from multidimensional and noisy data. In this regard, several studies have employed AI technology for predicting the volume of C&DW or MSW generated. For instance, an artificial neural network (ANN) algorithm has been applied to predict WG [11–19]. In addition, a support vector machine (SVM) algorithm has been utilized to develop a WG prediction model [12,13,18,20–28], and several studies have developed a WG prediction model using linear regression (LR) [11,13,29–36]. Moreover, a WG prediction model has been developed using a decision tree (DT) algorithm [3,37–40]. In particular, scholars prefer using standalone algorithms, e.g., ANN, SVM, LR, and DT algorithms, as AI models for WG prediction. Overall, the ANN and SVM algorithms are the most representative and frequently applied algorithms.

After Abbasi and Hanandeh (2016) developed a WG prediction model based on the k-nearest neighbor algorithm, several scholars have attempted to develop an AI model for WG prediction by applying a random forest (RF) algorithm [27,41–44]. The performance of the AI models rendered by the aforementioned standalone algorithms vary depending on the characteristics of the data, e.g., data size and input variable type, data processing method, and employed hyperparameters [45]. Generally, ANN and SVM algorithms are unsuitable for datasets in which the input-variable type is categorical data instead of numerical data [39,46]. In contrast, algorithms such as DT can be applied regardless of categorical data or numerical data, thereby yielding an acceptable performance. Furthermore, various performance results can be obtained for each study based on the selection of hyperparameters. Thus, the predictive performance of AI models developed using a standalone algorithm depends on the algorithm type, dataset characteristics (e.g., size or input-variable type), and selected hyperparameters. Accordingly, the predictive performance of AI models can be determined based on these influencing factors (e.g., algorithm type, data characteristics, and selected hyperparameters).

Although certain methods can be used to develop superior prediction models with improved prediction performance, these approaches pose considerable limitations. Recent research have focused on the development of hybrid AI models [13,14,18–21,24,47] to overcome the limitations associated with the existing standalone algorithms and augment the predictive performance of AI models. To this end, Abbasi et al. (2013, 2014) [20,21], Cai et al. (2020) [24], Dai et al. (2020) [47], Golbaz et al. (2019) [13], and Song et al. (2017) [18] developed hybrid models with improved predictive performance to predict C&DW and MSW generation by applying the following algorithms: the SVM algorithm, wavelet denoising method (WT), partial least-squares (PLS), long- and short-term memory (LSTM), fuzzy information granulation-genetic algorithm (FIG-GA), fuzzy, and gray model (GM). Furthermore, Liang et al. (2021) [14] and Soni et al. (2019) [19] improved the ANN model performance using Archimedes' optimization algorithm (AOA)–ANN and GA–ANN hybrid models, which enhanced the performance of the MSW prediction model as well. More importantly, the performance of existing hybrid models have improved in terms of their statistical metrics: root–mean–square (RMSE) values improved from 21% [21] to 48% [14], and coefficient of determination ($R^2$) increased from −6% [13] to 21% [19]. Similarly, other performance indicators, such as MSE, MAE, and MAPE, have demonstrated notable performance improvements in AI models. Overall, the characteristics common across most research on existing hybrid models include the application of the ANN and SVM algorithms with numerical data for the input-variable type of the dataset. Notably, ANN and SVM algorithms yield a superior performance in case of handling numerical data, which supports the stated trend.

This study aims to develop a hybrid DW model for predicting the volume of DW generation. Although existing studies primarily use numerical input data, this study proposes a method for developing a novel, hybrid predictive model that can deliver a

superior prediction performance in comparison to the DW prediction model applied with standalone algorithms for categorical input data. The following procedures were performed to fulfill the present research objective:

(1) Two distinct data-preprocessing methods were applied to construct the dataset. The first data-preprocessing method constructed a dataset by eliminating outliers, standardizing, and label-encoding categorical variables. The second method developed a dataset by reconstructing the numerical data using AE technology on the dataset obtained from the first method. After the application of the AE, various feature groups (i.e., number of features: 3, 6, 9, 15, 25, and 30) were created according to the representation size;

(2) The development of DW predictive models with standalone algorithms (i.e., ANN, SVR, and RF) and hybrid DW predictive models (i.e., AE–ANN, AE–SVR, and AE–RF) with respective algorithms and AE technology;

(3) The leave-one-out cross-validation (LOOCV) technique was utilized for model validation. The performances of various developed models were evaluated based on the statistical metrics of R, RMSE, $R^2$, and MAE;

(4) The performance results of DW predictive models and hybrid DW predictive models via standalone algorithms were compared and discussed. Finally, the optimal hybrid predictive model yielding the greatest performance improvement was proposed for DW generation, and the corresponding application method was discussed.

The remainder of the paper is organized as follows. The data and data processing methods used in this study, including the applied algorithms and the underlying reasons for their application along with the verification and evaluation method of the model, are introduced in Section 2. Thereafter, the performance results of the DW predictive models developed with standalone algorithms and the hybrid AE-based DW predictive models are comparatively analyzed in Section 3. In Section 4, the performance and applicability of the proposed hybrid model were compared with those of the existing models developed for predicting C&DW and MSW generation. Lastly, the major findings and limitations of this study along with the future scope of research are summarized in Section 5.

## 2. Methods and Materials

The data used in this study, data processing method, characteristics of the employed algorithms, and the validation and evaluation methods of the developed predictive models for predicting the demolition waste generation rate (DWGR) are described herein. In particular, the size and characteristics of the acquired data are detailed in Section 2.1; the data-preprocessing methods, including the categorical variables, are elaborated in Section 2.2, after which the application of the unsupervised and supervised machine learning (ML) algorithms is detailed in Section 2.3. Subsequently, the setup of the hyperparameters prior to the application of the algorithm is discussed in Section 2.4, and lastly, the verification and evaluation methods of the ML models developed for DWGR prediction are presented in Section 2.5. A schematic of the current research flow is illustrated in Figure 1.

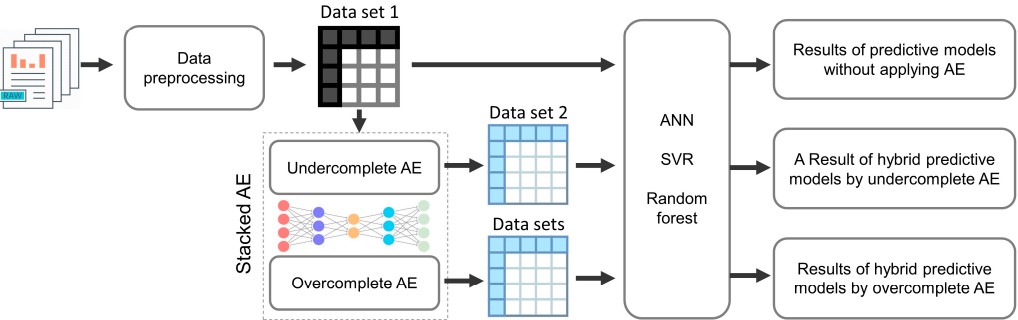

**Figure 1.** Flowchart of research steps performed in this study (undercomplete AE and overcomplete AE are detailed in Section 2.3).

*2.1. Data Source*

Building characteristics (e.g., floor area, usage, structure, and region) are the core influencing factors of DW generation [48]. Additionally, Banias et al. (2011) [49] and Cha et al. (2017) [37] considered the building usage (e.g., residential or commercial) as an important influencing factor for DW generation. Furthermore, Wang et al. (2021) [50] and Wu et al. (2021) [51] showed that the region in which the building is located influences the DW generation. Cha et al. (2017) [37] showed that wall type and roof type are also important factors influencing DW generation. Thus, to acquire information on the amount of decommissioning waste generated, the building characteristics (e.g., location, structure, usage, wall type, roof type, and gross floor area (GFA)) were obtained from a direct survey of 782 buildings prior to the demolition of buildings. The data included the records of dismantling waste discharge (kg/m$^2$) collected from demolition sites in redevelopment areas of two South Korean cities—Daegu and Busan. Thereafter, the dataset was constructed based on the information acquired from trucks disposing the demolition waste after building demolition, and the details of the demolition WG (kg) were obtained from the demolition company. A segment of the raw constructed dataset is presented in Table 1, wherein the size of the entire dataset was 782 rows × 7 columns. In Table 1, the building characteristics, e.g., the location, structure, usage, GFA, wall type, and roof type, represent the major factors affecting the DWGR. Therefore, we leveraged these six building features (i.e., location, structure, usage, GFA, wall type, and roof type) as variables of DWGR prediction. The relationship between the DWGR and these six building characteristics is expressed in Equation (1), and the DWGR is defined in Equation (2).

$$\text{DWGR} = f \text{ (location, structure, usage, GFA, wall type, roof type)}, \tag{1}$$

$$\text{DWGR}_i = \frac{\sum A \text{ of building}_i}{\text{GFA of building}_i}, \tag{2}$$

where DWGR$_i$ denotes the demolition waste generation rate (kg·m$^2$) of building i, A indicates the WG of building i (quantity; kg), and GFA symbolizes the gross floor area (m$^2$) of building i.

**Table 1.** Sample building input and output of raw data.

| Building Features | | | | | | | Output |
|---|---|---|---|---|---|---|---|
| **Bldg ID** | **Location** | **Structure** | **Usage** | **Wall Type** | **Roof Type** | **GFA (m$^2$)** | **Demolition Waste Generation (kg·m$^{-2}$)** |
| Bldg 1 | Project B | RC | Residential | Concrete | Slab | 289.50 | 1279.71 |
| Bldg 2 | Project A | RC | Residential | Brick | Slab | 114.20 | 2060.07 |
| Bldg 3 | Project A | RC | Residential | Brick | Slab | 100.45 | 3875.76 |
| Bldg 4 | Project A | RC | Residential | Brick | Slab | 100.45 | 1644.75 |
| Bldg 5 | Project A | RC | Residential | Brick | Slab | 197.68 | 1458.22 |
| Bldg 6 | Project A | RC | Residential | Brick | Slab | 190.36 | 2519.33 |
| Bldg 7 | Project A | RC | Residential | Brick | Slab | 114.80 | 2494.95 |
| Bldg 8 | Project A | RC | Residential | Brick | Slab | 118.41 | 3398.11 |
| Bldg 9 | Project A | RC | Residential | Brick | Slab | 47.11 | 1849.38 |
| Bldg 10 | Project A | RC | Residential | Brick | Slab | 106.45 | 2665.72 |
| Bldg 11 | Project A | RC | Residential | Brick | Slab | 87.53 | 2805.48 |
| Bldg 12 | Project A | RC | Residential | Brick | Slab | 82.40 | 3024.04 |
| Bldg 13 | Project A | RC | Residential | Brick | Slab | 95.15 | 6033.67 |
| Bldg 14 | Project A | RC | Residential | Brick | Slab | 51.11 | 2426.00 |
| Bldg 15 | Project A | RC | Residential | Brick | Slab | 149.51 | 1990.99 |
| . . . | . . . | . . . | . . . | . . . | . . . | . . . | . . . |
| Bldg 781 | Project C | Masonry | Residential | Block | Slate | 85.66 | 823.51 |
| Bldg 782 | Project B | Masonry | Commercial | Block | Slab and roofing tile | 94.44 | 1166.09 |

The basic statistical analysis of DWGR in terms of the building status and characteristics derived from the collected data is presented in Table 2, wherein the mean values of the DWGR characteristics observably varied with the location, usage, structure, wall type, and roof type of the buildings. In particular, the GFA and DWGR of most buildings were $\leq$300 (m$^2$) and $\leq$3000 (kg·m$^{-2}$), respectively, which can be attributed to the acquirement of data from redevelopment areas that predominantly housed old low-rise buildings.

**Table 2.** Building status and statistical analysis of raw data.

| Category | | Numbers | DWGR (kg·m$^2$) | | | |
|---|---|---|---|---|---|---|
| | | | Total | Min | Mean | Max |
| Location | Project A | 343 | 450,310 | 298 | 1313 | 6034 |
| | Project B | 356 | 485,037 | 83 | 1362 | 8574 |
| | Project C | 83 | 101,531 | 736 | 1223 | 1808 |
| Usage | Residential | 595 | 767,578 | 83 | 1290 | 8574 |
| | Residential and commercial | 172 | 251,381 | 418 | 1462 | 5718 |
| | Commercial | 15 | 19,510 | 607 | 1301 | 2474 |
| Structure | RC | 87 | 169,538 | 418 | 1949 | 6034 |
| | Masonry | 604 | 788,042 | 83 | 1305 | 8574 |
| | Wood | 91 | 80,889 | 298 | 889 | 2237 |
| Wall type | Concrete | 9 | 10,357 | 871 | 1151 | 4696 |
| | Brick | 236 | 391,259 | 252 | 1658 | 6034 |
| | Block | 500 | 596,799 | 83 | 1194 | 8574 |
| | Mud plastered and mortar | 37 | 40,056 | 517 | 1083 | 2591 |
| Roof type | Slab | 289 | 479,356 | 252 | 1659 | 6034 |
| | Slab and roofing tile | 33 | 38,877 | 252 | 1178 | 1808 |
| | Slate | 178 | 227,923 | 306 | 1280 | 8574 |
| | Roofing tile | 282 | 292,314 | 83 | 1037 | 2527 |

### 2.2. Data Preprocessing and Dataset Size

As the construction of a stable dataset is a prerequisite for improving the performance of ML predictive models, we performed data preprocessing on the acquired dataset. The categorical input variables were converted into numerical variables by preprocessing the data as follows: encoding, outlier elimination, and standardization. First, encoding was performed to convert the data into vectors of real numbers, which would act as the input variables for the AE. As AEs contain neural network structures that operate on real vectors [52], the categorical data must be converted into a real vector to enable the application of a neural network on qualitative data [53]. This study implemented label encoding to convert the categorical variables into real vectors; the label encoding regime for each categorical variable is listed in Table 3. The data-encoding process was followed by outlier elimination according to Equation (3), after which the size of the processed dataset was reduced to 690 rows × 7 columns. In particular, this encoded and processed dataset was employed for developing the ML model. Thereafter, data standardization was performed according to Equation (4) to construct a dataset of the same scale unit.

$$\text{Q1} - 1.5 \times \text{IQR} < \text{selecting data} < \text{Q3} + 1.5 \times \text{IQR}, \tag{3}$$

where IQR denotes the interquartile range, derived as Q3 − Q1; Q denotes the quartile, Q1 indicates the 25th percentile, and Q3 denotes the 75th percentile.

$$x_{\text{standardization}} = \frac{x - \bar{x}}{\sigma}, \tag{4}$$

where x, $\bar{x}$, and $\sigma$ represent the element, mean, and standard deviation of the data, respectively.

**Table 3.** Label encoding for numerical data conversion of categorical variables in this study.

| Categorical Variable | | Numerical Value Assigned by Label Encoding |
|---|---|---|
| Location | Location_project A | 0 |
| | Location_project B | 1 |
| | Location_project C | 2 |
| Structure | Structure_RC | 0 |
| | Structure_masonry | 1 |
| | Structure_wood | 2 |
| Usage | Usage_ residential | 0 |
| | Usage_residential & commercial | 1 |
| | Usage_ commercial | 2 |
| Wall type | Wall type_concrete | 0 |
| | Wall type_brick | 1 |
| | Wall type_block | 2 |
| | Wall type_ mud plastered and mortar | 3 |
| Roof type | Roof type_slab | 0 |
| | Roof type_slab and roofing tile | 1 |
| | Roof type_slate | 2 |
| | Roof type_roofing tile | 3 |

### *2.3. Application of ML Algorithms*

The inherent properties of the AE algorithm, the reasons for its implementation in this study, and its overall applications are described herein. In addition, we examined the characteristics of the supervised learning algorithms adopted for developing the DWGR prediction model and discussed the reasons for its adoption.

#### 2.3.1. Autoencoder (Unsupervised Learning)

AE is an ANN-based unsupervised ML algorithm [54] that utilizes a neural network to reconstruct an output value equal to an arbitrary input value [55]. In particular, the basic AE contains a symmetrical structure comprising two functional segments—encoder and decoder—and three layers: input layer, hidden layer, and output layer. The encoder transforms the original input data (X) into a lower-dimensional layer, called the compressed representation (also known as feature or latent vector). Additionally, the decoder decompresses the representation into new input data (X′) reconstructed according to the relationship between the input variables [56]. Thus, the features of the input values regenerated by the AE exhibit numerical differences. This AE characteristic is beneficial for supplementing the characteristics of numerical values simply converted via label encoding into ordinal variables. This is because the categorical variables converted via label encoding are arbitrarily assigned integer values, and information cannot be used because of the numerical difference between the encoded values [53]. As the model performance can be expectedly improved by considering the numerical difference characteristics to these categorical variables, this study implemented an AE to convert the categorical input variables into input variables with numerical information.

In the AE, the size of the hidden layer determines the size of the representation (latent vector or feature) [57]. This algorithmic property of the AE can be utilized for feature engineering. Specifically, the number of features depends on the size of the hidden layer. Although AEs can improve the performance of regression and classification tasks through lower-dimensional representations [52], the dimensionality in this case should be reduced to accommodate the substantially high number of features. Conversely, in the case of fewer features, the number of features can be increased using an overcomplete AE on the original dataset, i.e., an AE architecture with a hidden layer dimensionally larger than the input layer [58]. Nonetheless, an overcomplete representation is advantageous because it increases the stability of the representation [59]. However, as reported by Fan et al.

(2019) [60] and Meyer (2015) [61], if the hidden layer size is larger than the input layer size in a basic AE, a problem may occur in the generalization of the mode. As observed in stacked AEs, the generalization issue can be resolved by incorporating a hidden layer [62]. In a stacked AE, the pre-training is performed by considering a single layer at a given instant. The layers of the stacked AE are trained to individually minimize the errors arising from the input reconstruction of the layers [63]. In particular, the pre-training aids generalization by ensuring that the learned information is extracted from the input [62,63].

Therefore, the stacked AE architecture was applied in this study to implement a hidden layer that was larger than the input layer, as illustrated in Figure 2. Moreover, we constructed a dataset in which the number of features was adjusted according to the undercomplete and overcomplete AE architectures by altering the representation size of the hidden layer. As indicated in Figure 2, the architectures of an undercomplete AE (feature representation size = 3) and six overcomplete AEs (feature representation sizes = 6, 9, 15, 20, 25, and 30) were applied to construct the feature sets with data sizes of 3, 6, 9, 15, 20, 25, and 30 columns.

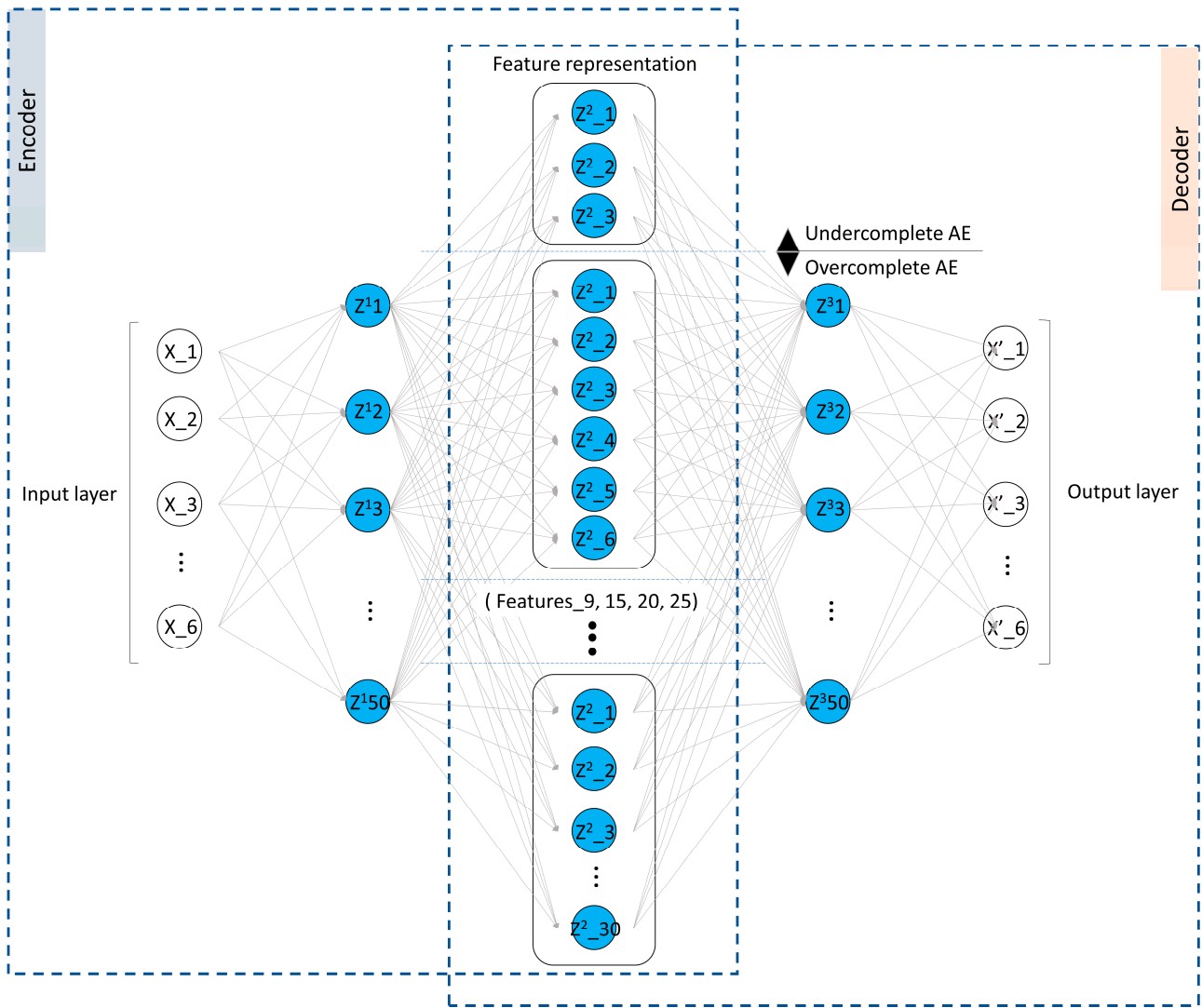

**Figure 2.** Stacked AE architecture based on applied representation size.

In general, the AE minimizes the distance between the input and output by maximally recovering the information from the original input [64]. To this end, the AE model uses loss functions to recreate the features and updates the weight parameters to obtain more efficient results, as well as reduce the likelihood of errors. In addition to minimizing the

loss, the AE converges the value of X′ to that of X. Thus, the appropriate transformation of the input variable via the AE can be determined from the loss result. In particular, the loss function based on the reconstruction error is expressed in Equation (5). According to the loss result, the data conversion conducted in this study was assessed to be appropriate.

$$L(x, x') = \frac{1}{n} \sum_{i=1}^{n} (x - x')^2, \tag{5}$$

where x and x′ indicate denotes the output and input of the decoder, respectively.

### 2.3.2. Supervised Learning Techniques Used for DWGR Estimation

This study applied supervised learning algorithms to various sets of features constructed using the AE, i.e., subsets comprising 3, 6, 9, 15, 20, 25, and 30 features. In particular, the SVR, ANN, and RF were selected as the learning algorithms. The SVR and ANN algorithms have been widely applied in the field of waste prediction [45,65,66]. Moreover, RF is an ML algorithm with excellent predictive performance and is considered one of the ten best classifiers [67]. In particular, RF can utilize both categorical and continuous variables and is useful for comparing performance variations with the application of AE on SVR and ANN. The details of the supervised learning techniques employed in this study are described below.

### Random Forest

Initially proposed by Breiman (2001) [68], RF is a representative ensemble technique based on bagging, which is used to perform bootstrap sampling. Overall, RF is considered one of the most powerful ML algorithms. In principle, RF creates a tree (called weak learner) for each subset by extracting multiple subsets (bootstrap sampling) from the original dataset. The final prediction determines the strong learner based on the majority of votes obtained from the results of each tree. Through this process, RF can prevent overfitting as the number of trees increases and is minimally influenced by outliers. Even for an unbalanced class, this classification algorithm offers superior predictive performance compared to other machine learning algorithms [67]. Recently, certain researchers used RF to predict the amount of waste generated in the fields of C&D and solid WM; Cha et al. (2020, 2021) [41,42] applied RF to estimate the DWGR for diverse types of waste generated during the building–dismantling process, whereas Kumar et al. (2018) [27] leveraged RF to predict the plastic WG rate. Additionally, Dissanayaka and Vasanthapriyan (2019) [43] employed RF to predict MSW generation, and Namoun et al. (2022) [69] used RF to predict household solid WG. Furthermore, this algorithm was implemented by Rosecký et al. (2021) [39] to predict MSW generation at the regional level.

### Support Vector Regression

The working principle of SVR is stated as follows: a linear decision function is constructed in the feature space by mapping the input data to the feature space using a nonlinear map. The principal aim of SVR is to determine the optimal decision function. Thereafter, using kernels, SVR nonlinearly maps a linear decision function of the feature space to the original space [28]. The SVR model intends to overcome the fundamental drawback of parametric regression. Overall, it is a novel and powerful ML technique based on statistical learning theory and adheres to the principle of structural risk minimization, i.e., it aims to minimize the upper bound of the generalization error instead of minimizing the training error [25,28,70]. As reported, the SVR model performs reliably in solving problems with small samples, nonlinearities, and high-dimensional characteristics [71]. Coupled with ANN, the SVR model has been effectively utilized for AI models in various fields, and several researchers have applied it in WG-related fields as well. In this regard, Abbasi et al. (2014, 2013) [20,21], Abbasi and Hanandeh (2016) [22], Abunama et al. (2019) [23], Dai et al. (2020) [47], Golbaz et al. (2019) [13], Graus et al. (2018) [26], Kumar et al. (2018) [27], and

Song et al. (2017) [18] used SVR to predict MSW generation, whereas Cai et al. (2020) [24] and Cha et al. (2022) [12] employed SVR to predict C&DW and DW generation.

Artificial Neural Networks

The ANN theory was initially proposed by McCulloch and Pitts (1943). According to the signal transmission modes, ANNs are classified as feedforward and feedback neural networks. Owing to their simpler and superior performance over feedback neural networks, feedforward neural networks have been widely used in the field of WM [68]. As reported in the literature, the multilayer perceptron (MLP) is a prominent feedforward ANN architecture that is employed for forecasting problems [72,73]. The basic structure of an MLP contains three layers: input, hidden, and output layers, and the nonlinear transfer function comprising multiple layers of neurons enables the learning of nonlinear and linear relationships between input and output neurons. These neurons are connected to the adjacent layer and hidden layer, and the interconnections of all neurons contain weights indicating the strength. The number of input neurons—identical to the number of input variables—is responsible for receiving external information. In the hidden layer, the sum of weights is passed through an activation function that determines the relationship between the input and output. Subsequently, the output of the hidden layer is inputted to the output layer, and the weights are calculated and transformed via the linear activation function of the output layer [29,74]. The ANN algorithm is the most frequently used algorithm for AI models in the field of WM [45]. In relation to WG, ANNs have been utilized by Golbaz et al. (2019) [13] to predict solid WG from hospitals, by Liang et al. (2021) [14], Shamshiry et al. (2014) [17], and Soni et al. (2019) [19] to predict MSW generation, and by Cha et al. (2022) [12] and Song et al. (2017) [18] to predict C&DW generation.

### 2.4. Application of Algorithms and Hyperparameter-Tuning

Hyperparameters significantly influence the predictive performance, robustness, and generalization ability of a model. Therefore, prior to the application of AE, we tuned the hyperparameters to derive the optimal performance model for each algorithm (i.e., RF, ANN-MLP, and SVR). For instance, we adjusted the ANN models according to the number of hidden layers and neurons, and the ANN (MLP) models were tested with 1–4 hidden layers and 20, 30, and 40 neurons for each hidden layer. Based on the test results, we selected the ANN-MLP model with the structure of input layer—first hidden layer (10 neurons); second layer (30 neurons)—output layer and two hidden layers, featuring 10 neurons in the first hidden layer and 30 neurons in the second hidden layer.

For the SVR model, hyperparameters such as the kernel, kernel coefficient ($\gamma$), and penalty parameter of the error term (C) were adjusted in this study. In addition, a radial basis function kernel, widely used in regression problems, was considered for the kernel. This strategy was selected considering that it can nonlinearly map samples to a high-dimensional space and can easily handle nonlinear relationships between class labels and properties [75]. As for the kernel coefficient, $\gamma = 1/6$ was applied according to $\gamma = 1/K$ (where K denotes the number of features), as proposed by Chang and Lin (2011) [76]. For the penalty parameter of the error term (C), the optimal value was searched for values ranging from $10^{-5}$ to 101, and the value of C = $10^{-3}$ yielded the optimal performance results.

For the RF model, the optimal predictive performance was derived by adjusting the number of trees and features. The number of trees was varied in increments of 50 units ranging from 100 to 500 and tested with submodels containing 3, 4, 5, and 6 variables. The test results revealed that the RF model delivered the optimal results with 450 trees and 6 variables.

### 2.5. Model Evaluation

2.5.1. Model Validation

In this study, the proposed model was validated using LOOCV, which is a special case of k-fold cross-validation technique. LOOCV is regarded a suitable validation method

for small sample sizes [77,78]. Accordingly, several studies have adopted LOOCV to evaluate the performance of algorithms handling a small number of instances in the dataset [8,79]. In principle, LOOCV utilizes all samples as the test-and-training data to secure adequate training and validation sets. Compared with the 10-fold or k-fold cross-validation, LOOCV is advantageous for obtaining stable results with small target datasets [41,42,80,81]. Therefore, considering the size of the dataset in this study, LOOCV was applied as a model validation method.

### 2.5.2. Performance Measures

The performance of the DWGR predictive models developed in this study were evaluated based on statistical metrics such as the MAE (Equation (6)), RMSE (Equation (7)), $R^2$ (Equation (8)), and R (Equation (9)). Generally, a satisfied model yields high $R^2$ and R values and low MAE and RMSE values.

$$\text{MAE} = \frac{\sum_{i=1}^{n}|y_i - x_i|}{n} \tag{6}$$

$$\text{RMSE} = \sqrt{\sum_{i=1}^{n}\frac{(y_i - x_i)^2}{n}} \tag{7}$$

$$R^2 = 1 - \frac{\sum_{i=1}^{n}(y_i - x_i)^2}{\sum_{i=1}^{n}(y_i - \overline{x}_i)^2} \tag{8}$$

$$R = \frac{\sum_{i=1}^{n}\left(x_i - \overline{x}_i\right)\left(y_i - \overline{y}_i\right)}{\sqrt{\sum_{i=1}^{n}\left(x_i - \overline{x}_i\right)^2}\sqrt{\sum_{i=1}^{n}\left(y_i - \overline{y}_i\right)^2}} \tag{9}$$

where $x_i$ denotes the observed quantity of generated DW, $y_i$ represents the predicted quantity of the generated DW, $\overline{x}_i$ denotes the mean observed quantity of generated DW, $\overline{y}_i$ indicates the average predicted quantity of generated DW, and n denotes the number of samples.

### 3. Results

#### 3.1. Learning Validity Assessment of the Stacked AE Utilized in this Study

The loss function values derived during the learning process are plotted in Figure 3. In the current case of the representation applied with six features, the validation loss approximated the training loss at nearly epoch 20, and thereafter, converged to 0. The validation loss results of the six-feature representation revealed that overfitting did not occur in the applied AE model. For the remaining feature representations (i.e., 3, 9, 12, 15, 20, 25 features; refer to Supplementary Material, Figure S1), the validation loss was larger than the train loss, ranging from approximately epochs 15–20. However, in all feature representations beyond this range, it yielded stable results converging to 0. As observed from these loss results (refer to Figures 3 and S1), overfitting did not occur in all feature representations and the validation loss exhibited a pronounced declivity. In addition, the stacked AE architecture evidently facilitated the stable conversion of the categorical input variable data by considering the numerical information relationship between the input variables, as it maintained the existing data characteristics. The distribution of data values converted from categorical variables into numerical variables through the stacked AE is depicted in Figure S2 (refer to Supplementary Material).

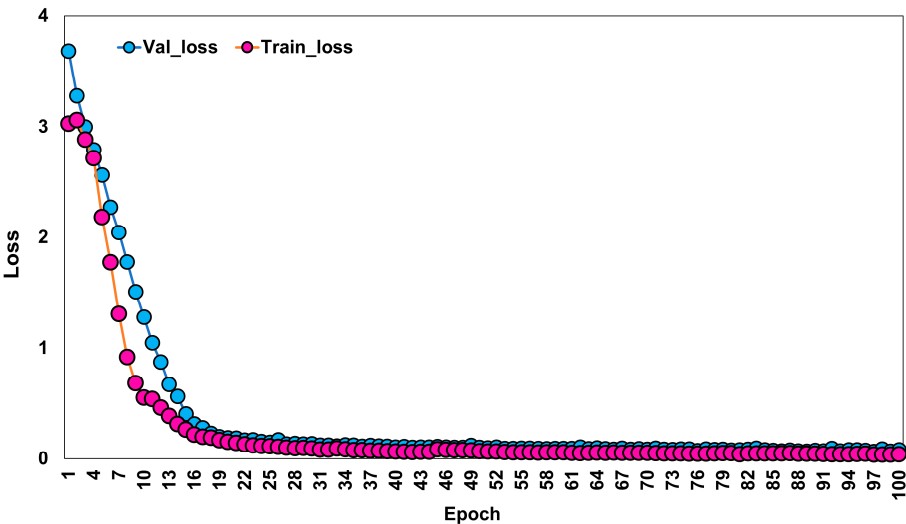

**Figure 3.** Training and validation loss on six-feature representation by AE.

### 3.2. Comparison of Performance Results and Improvement of Models

The performance results of the predictive models based on standalone algorithms (i.e., ANN-MLP, SVR, and RF) and hybrid models combined with AE for the DWGR prediction are comparatively presented in Figure 4. First, as demonstrated by the results of ANN-MLP (Figures 4 and 5a), the performance of the model with AE implementation yielded significantly improved results for all performance indicators (i.e., MAE, RMSE, $R^2$, and R). In terms of model stability, the MAE and RMSE of all AE–ANN (MLP) models were significantly stabilized below 212.133 and 274.371 (i.e., MAE and RMSE values of worst-performing AE (three features)–ANN (MLP) model), respectively, compared to the ANN (MLP) model (MAE: 356.697; RMSE: 316.186). In particular, the AE (25 features)–ANN (MLP) model yielded the lowest values of the MAE and RMSE at 182.105 and 230.819, respectively. In addition, the $R^2$ and R values were significantly improved in comparison to those of the ANN ($R^2$: 0.458; R: 0.676), and the $R^2$ and R of the AE (25 features)–ANN (MLP) model were 0.680 and 0.825, respectively, which were superior to those of the remaining AEs–ANN (MLP) models.

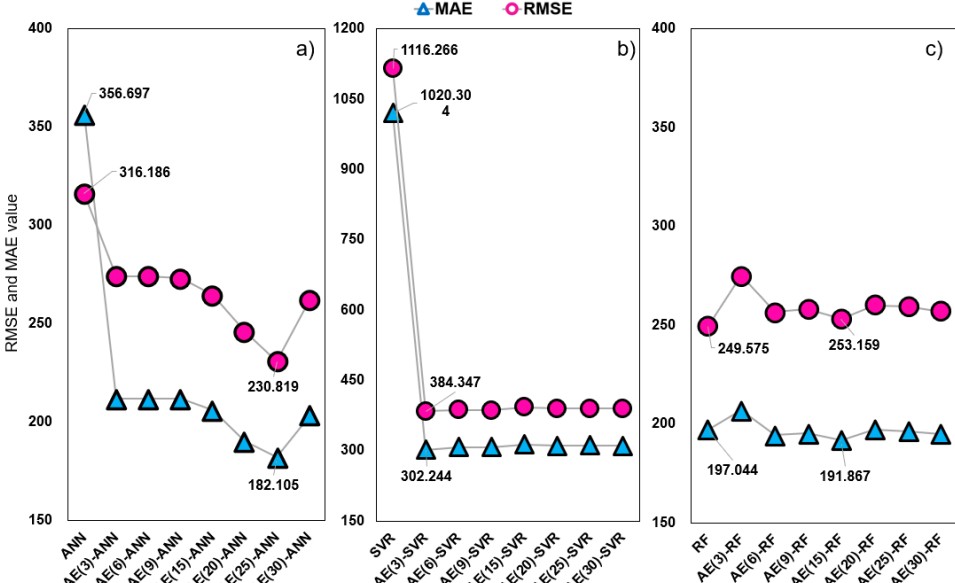

**Figure 4.** Performance comparison of RMSE and MAE of hybrid models and non-hybrid models. (**a**) ANN (MLP), (**b**) SVR, and (**c**) RF.

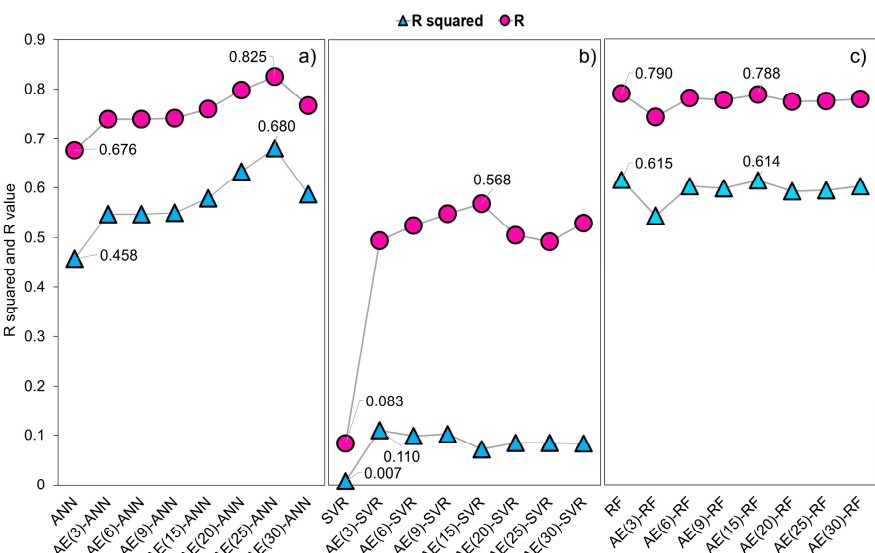

**Figure 5.** Performance comparison of $R^2$ and R of hybrid models and non-hybrid models. (**a**) ANN (MLP), (**b**) SVR, and (**c**) RF.

In the SVR results depicted in Figures 4 and 5b, the MAE and RMSE of all AE–SVR models were significantly more stable in comparison to those of the SVR models (MAE: 1020.304; RMSE: 1116.266). Specifically, the MAE and RMSE of the AE (three features)–SVR model were 302.244 and 384.347, respectively, thereby demonstrating optimal results with no significant variation from the MAE and RMSE results of remaining AE–SVR models. Moreover, as noted from the R values, the performance of AE–SVR models was significantly improved compared to that of the SVR model, wherein the AE (15)–SVR model (R: 0.568) yielded the optimal R value. However, in terms of $R^2$ values, all AE–SVR models delivered improved their performance over the existing SVR model ($R^2$: 0.007). In contrast, the R of the AEs–SVR models ranged from 0.071–0.110 and suggested the inferior accuracy of the AEs–SVR models, which signifies the inapplicability of these models for prediction purposes.

The performance results of the RF and AE–RF models are plotted in Figures 4 and 5c, which indicate distinct patterns compared to those of the AE–ANN (MLP) and AE–SVR models. For the prediction model applying solely the RF algorithm, the MAE and RMSE results were 197.004 and 249.597, respectively, whereas those of the AE–RF models were 191.867–206.940 and 253.159–275.385, respectively. In terms of stability, the MAE of certain AE–RF models was superior to that of RF models. However, their RMSE results were inferior to those of RF models in all cases. In terms of $R^2$ and R values, the performance of all the AE–RF models was marginally worse than the RF model. These results indicated negligible improvement in the performance of AE–RF models. Essentially, as the RF algorithm can handle all types of categorical and numerical variables, the application of variable conversion technology such as AEs can be evidently ineffective. However, as depicted in Figures 4 and 5, a significant performance improvement was observed for the ANN and SVR algorithms. The ANN model yielded a significant improvement in all the performance indicators (MAE, RMSE, $R^2$, and R), and in particular, the predictive performance of the AE (25)–ANN (MLP) model was superior to that of the RF model. Furthermore, the SVR model displayed a significant improvement in the performance indicators (MAE, RMSE, and R), and in terms of $R^2$, the resulting performance improvement was inadequate.

As indicated by the aforementioned performance index results, the AE (25)–ANN (MLP) model delivered the most outstanding performance for predicting DW generation. The degree of performance improvement obtained by applying AE can be clearly confirmed by comparing the correlation results between the observed and predictive values of the

ANN (MLP) model and AE (25 features)–ANN (MLP) model (Figure 6a,b), and the results of the predictive model (Figure 7). As indicated in Figure 6a,b, the observed and predictive values of the AE (25 features)–ANN (MLP) model, compared to the ANN (MLP) model, were more closely distributed along the line with a correlation coefficient of 1. Moreover, as depicted in Figure 7, the predicted values of the AE (25 features)–ANN (MLP) model more closely approximated the trend of the observed values than the ANN (MLP) model. The mean of the observed values was 1165.04 kg·m$^{-2}$, and the mean of the predictive values of the ANN (MLP) and AE (25 features)–ANN (MLP) models were 1157.656 kg·m$^{-2}$ and 1162.437 kg·m$^{-2}$, respectively. These results confirmed the significant improvement of the predictive performance of the AE (25 features)–ANN (MLP) model. Moreover, the utilization of AE can be considered an advantageous new method for the development of predictive models with excellent performance on datasets containing categorical variables.

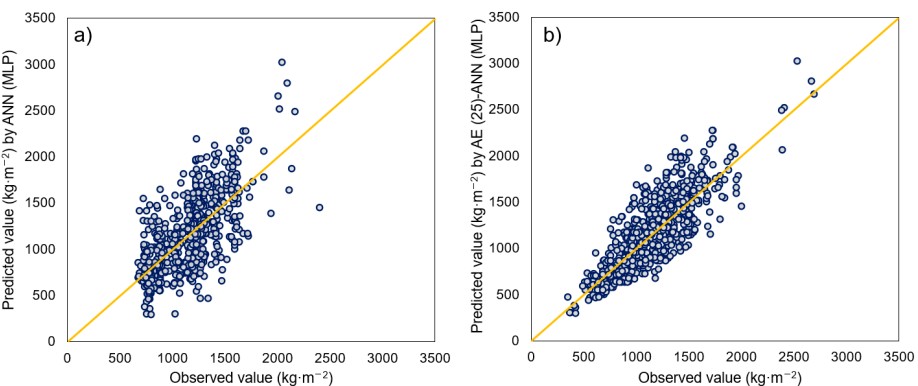

**Figure 6.** Correlation comparison between predicted and observed values. (**a**) ANN (MLP) and (**b**) AE (25)−ANN (MLP).

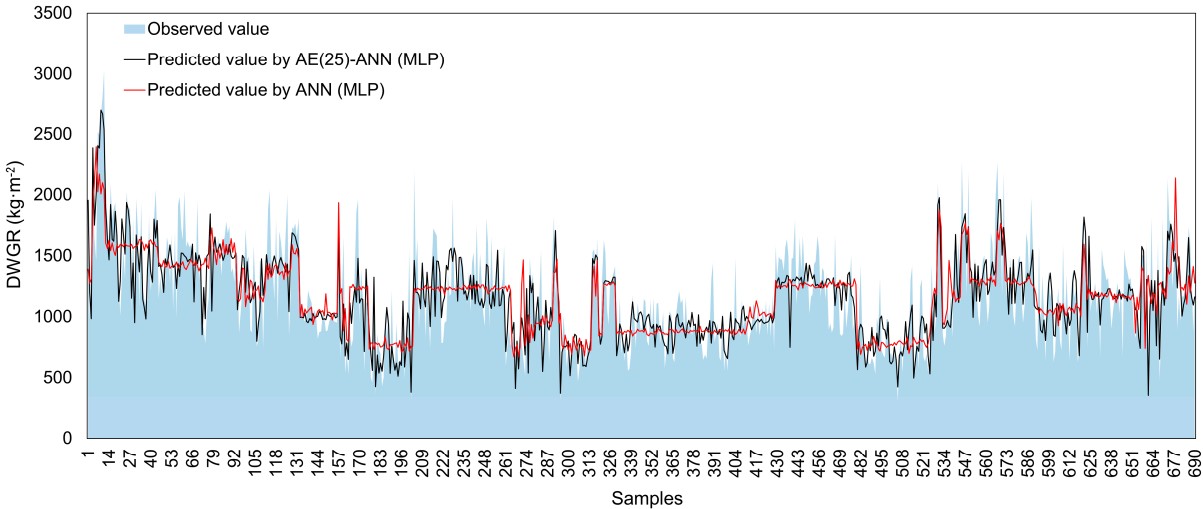

**Figure 7.** Comparison results of observed and predicted values.

## 4. Discussion and Recommendations

To more accurately estimate the generation of C&DW during building demolition, this study aimed to develop a hybrid predictive model that can improve the predictive performance of a dataset comprising categorical data. To date, numerous studies on hybrid model development have attempted to improve the performance of the WG prediction model in the fields of C&DW and MSW [12–14,18–21,24,47]. As listed in Table 4, the existing hybrid ML models aimed to predict C&DW and MSW generation at various estimation levels such as building, district, city, and region. Commonly, these studies applied the ANN and SVR techniques as ML algorithms, primarily using numerical data as the input variables.

**Table 4.** Comparison of performance results of non-hybrid and hybrid models for prediction of C&D and solid WG.

| Study | Estimation Level | Waste Type | Input Variable Data Composition (Number of Input Variables or Characteristics of Data) | Model Type | Performance of Predictive Model | | | | | | |
|---|---|---|---|---|---|---|---|---|---|---|---|
| | | | | | RMSE | MSE | MAE | MAPE | $R^2$ | R | Error Rate (%) |
| This study | Building | DW | Numerical (1); categorical (5) | ANN (MLP) | 316.186 | | 356.697 | | 0.458 | 0.676 | |
| | | | | AE (25 features)–ANN (MLP) | 230.819 | | 128.105 | | 0.680 | 0.825 | |
| | | | | SVR | 1020.304 | | 1116.266 | | 0.007 | 0.083 | |
| | | | | AE (3 features)–SVR | 302.224 | | 384.347 | | 0.110 | 0.494 | |
| Abbasi et al., 2013 [20] | City | MSW | Numerical (time series data) | SVM | 2070 | | | | | 0.761 | |
| | | | | PL–SVM | 1541 | | | | | 0.869 | |
| Abbasi et al., 2014 [21] | City | MSW | Numerical (time series data) | SVM | 814–3268 | | | | | 0.702–0.756 | |
| | | | | WT-SVM | 639–2283 | | | | | 0.813–0.887 | |
| Song et al., 2017 [18] | Regional | C&D waste | Numerical (time series data) | GM | | | | | | | 21 |
| | | | | GM -SVR | | | | | | | 4.6 |
| Golbaz et al., 2019 [13] | Building | MSW (hospital solid waste) | Numerical (7); categorical (1) | SVM | | 0.001–0.003 | | | | 0.79–0.98 | |
| | | | | F–SVM | | 0.001–0.002 | | | | 0.79–0.92 | |
| Soni et al., 2019 [19] | City | MSW | Numerical (4) | ANN | 165.5 | | | | | 0.72 | |
| | | | | GA–ANN | 95.7 | | | | | 0.87 | |
| Cai et al., 2020 [24] | Regional | C&D waste | Numerical (time series data) | SVR | 50.19 | | | 17.29 | | | |
| | | | | LSTM–SVR | 29.04 | | | 10.02 | | | |
| Dai et al., 2020 [47] | District | MSW | Numerical (3; time series data) | SVR | | 34.725 | | 14.434 | 0.8376 | | |
| | | | | FIG–GA–SVM | | 5.703 | | 2.012 | 0.9845 | | |
| Liang et al., 2021 [14] | City | MSW | Numerical (8) | ANN | 11.23 | | 10.29 | | | 0.76 | |
| | | | | AOA–ANN | 5.89 | | 6.21 | | | 0.88 | |

For instance, Abbasi et al. (2013) [20] developed a PLS–SVM hybrid model to improve the performance of the SVM model based on daily solid WG-related time-series data from 2004 to 2005 of Tehran city. Moreover, Abbasi et al. (2014) [21] attempted to enhance the performance of the SVM model using the WT method on the MSW time-series data corresponding to Tehran city. Furthermore, Song et al. (2017) [18] developed a GM–SVR hybrid model to predict C&DW generation in 31 provinces of China, and reported a significant improvement in its performance.

Additionally, Golbaz et al. (2019) [13] leveraged multiple ML models, such as multilinear regression, ANN, SVM, least squares–SVM, and fuzzy logic (F)–SVM, to predict solid WG from hospitals, among which the F–SVM hybrid model yielded the best performance. Furthermore, Soni et al. (2019) [19] developed a GA–ANN model to predict MSW generation. The GA–ANN model yielded performance improvements of 42% and 21% compared to the ANN model, respectively. More recently, Cai et al. (2020) [24] developed hybrid models using the statistical analysis models of autoregressive integrated moving average, SVR, back-propagation neural network, and LSTM algorithms, among which the LSTM–SVR model delivered the best performance.

Dai et al. (2020) [47] developed a FIG–GA–SVR model to predict district-level MSW generation (Huangshi city, Hubei Province, China). The FIG–GA–SVR model produced performance improvements of 84% and 86% in terms of MSE and MAPE compared to the SVR model, respectively. Additionally, Liang et al. conducted a study on hybrid models, combining algorithms such as ANN, GA, particle swarm optimization, a sine–cosine algorithm, and AOA to predict MSW generation in major cities of Iran. Among the hybrid models, the AOA–ANN model yielded the best predictive performance.

The degree of performance improvement of the hybrid AI models developed for predicting C&D and solid WG was 21%–48% in terms of RMSE (derived from the results of [14,19–21,24]), 40% in relation to MAE (deduced from the results of [14]), and 0–21% in $R^2$ values (based on the results of [13,14,19–21,24]). On the other hand, the AE (25 features)–ANN(MLP) model developed in this study yielded performance improvements of 49%, 27%, 49%, and 22% in terms of MAE, RMSE, $R^2$, and R, respectively (listed in Table 4), compared to the non-hybrid ANN model. Moreover, compared to the non-hybrid SVR model, the AE (three features)–SVR model yielded an improvement of 70% and 66% in terms of MAE and RMSE, respectively (refer to Table 4). Therefore, the application of the AE technique to the categorical variables in this study significantly influenced the stabilization and accuracy of the model, which resulted in superior performance improvements compared to the existing hybrid models. As summarized in Table 4, unlike previous studies that developed hybrid models for numerical data, this study explored a new direction with the proposed hybrid model to significantly improve the prediction performance for categorical data.

As depicted in Figures 4 and 5, the prediction performance of the RF algorithm for categorical data is superior to those of the ANN and SVR algorithms. In general, the RF algorithms are considered adequately robust to ensure an optimal prediction performance [66]. Accordingly, these algorithms have been widely used in several fields owing to their excellent performance, as well as fast and efficient training process [82]. Therefore, developing an ML regression model with a predictive performance superior to that of the RF model is a challenging task. However, the predictive model proposed in this study combined the ANN and SVR algorithms with AE technology and used categorical variables to deliver a superior performance compared to the RF model. Therefore, this study presents a new strategy to overcome the limitations emerging from the characteristics of the variable type. As demonstrated, the development of hybrid model via AEs can be advantageous toward achieving excellent predictive models with superior performance.

This strategy is beneficial in the case of applying ML algorithms that require numerical data, such as ANN and SVR, or in case (1) The dataset contains more categorical data than numerical data; or (2) The implemented ML algorithm is applicable regardless of the input variable type, such as RF (e.g., decision tree, extra classifier tree, Xgboosting, and gradient-boosting machine). In particular, the proposed strategy will be of great interest to demolition

companies because it can facilitate the development of a DW waste-prediction model in a limited-data environment (e.g., lack of numerical input information for buildings subjected to demolition and building characteristics data comprising simple nominal variables). Moreover, the proposed method can be combined with various ML algorithms to improve the predictive performance of models for C&DW and MSW management. Furthermore, the application of the current methodology in a similar data environment will be advantageous for developing models with excellent predictive performance in several other fields as well.

## 5. Conclusions

This study developed a novel hybrid AI model using standalone algorithms with AE technology to predict DWG from the demolition of buildings in redevelopment areas in South Korea. This study reports novel research findings relevant to the field of C&DW and MSW management. The performance of the ANN-MLP and SVR models was improved using categorical data, and a hybrid DW predictive model was developed by applying AE. In particular, compared to the ANN (MLP) model, the AE (25)–ANN (MLP) model improved performance in terms of MAE, RMSE, $R^2$, and R, thereby significantly influencing the model's stability and accuracy. The mean of the observed values was 1165.04 kg·m$^{-2}$, and those of the predictive values obtained by the ANN (MLP) and AE (25)–ANN (MLP) models were 1157.656 kg·m$^{-2}$ and 1162.437 kg·m$^{-2}$, respectively. In addition, the application of AE technology to the SVR algorithm considerably enhanced the prediction stability of the AI model based on the significantly improved results in terms of MAE and RMSE.

Notably, the potential of performance improvement with AE will be insignificant for algorithms (e.g., RF) that can handle both categorical and numerical data as an input variable type. However, as the AE (25)–ANN (MLP) predictive model demonstrated a superior performance compared to the RF predictive model, this method is considered a novel and advantageous approach for developing a DW predictive model because it produced an excellent predictive performance for datasets comprising categorical data. More importantly, the present findings are crucial for surpassing the limitations of the feasible ML algorithms that rely on data characteristics and developing various AI models with superior predictive performance. Therefore, the results of this study can help develop ML models with better performance in various fields. In addition, the results of this study are expected to be useful for decision-making by related industry officials or policymakers by providing more accurate DW generation prediction information.

However, as this study was limited to the use of ANN, SVR, and RF algorithms, more diverse algorithms should be considered in the future, and further research should be conducted on various datasets. For example, in the future, it is necessary to conduct research on the development of ML models using AE for datasets other than the categorical input variables used in this study. In addition, efforts are required to find an optimal DWGR ML model by applying additional algorithms.

**Supplementary Materials:** The following supporting information can be downloaded at: https://www.mdpi.com/article/10.3390/su15043691/s1, Figure S1: Loss results according to feature representation size of AE: 3, 9, 15, 20, 25, and 30 features arranged from (a–g), respectively; Figure S2: Distribution of data values by feature representation size: 3, 6, 9, 15, 20, 25, and 30 features arranged from (a–g), respectively.

**Author Contributions:** G.-W.C. and Y.-C.K.: Conceptualization, methodology, validation, and supervision. G.-W.C. and Y.-C.K.: Writing—original draft preparation. G.-W.C.: Formal analysis. G.-W.C. and Y.-C.K.: Resources. G.-W.C., Y.-C.K. and W.-H.H.: Writing—review and editing and funding acquisition. All authors have read and agreed to the published version of the manuscript.

**Funding:** This work was supported in part by the National Research Foundation of Korea (NRF), grant funded by the South Korean government (MSIT) (NRF-2019R1A2C1088446). This work was supported by the National Research Foundation of Korea (NRF), grant funded by the South Korean government (MSIT) (No. NRF-2020R1C1C1009061).

**Institutional Review Board Statement:** Not applicable.

**Informed Consent Statement:** Not applicable.

**Data Availability Statement:** All data included in this study are available upon request by contact with the corresponding author.

**Conflicts of Interest:** The authors declare no conflict of interest.

**Abbreviations**

| | |
|---|---|
| AI | Artificial Intelligence |
| AE | Autoencoder |
| ANN | Artificial Neural Network |
| C&DW | Waste, Construction, and Demolition Waste |
| CV | Cross Validation |
| DW | Demolition Waste |
| DWGR | Demolition Waste Generation Rate |
| GFA | Gross Floor Area |
| LOOCV | Leave-One-Out Cross Validation |
| ML | Machine Learning |
| MSW | Municipal Solid Waste |
| MAE | Mean Absolute Error |
| MLP | Multilayer Perceptron |
| MSE | Mean Squared Error |
| R | Pearson's Correlation Coefficient |
| $R^2$ | Coefficient of Determination |
| RMSE | Root–Mean–Square Error |
| RF | Random Forest |
| SVR | Support Vector Regression |
| WG | Waste Generation |

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
