# Peer review of "Performance Improvement of Machine Learning Model Using Autoencoder to Predict Demolition Waste Generation Rate"

_sustainability, doi:10.3390/su15043691_

Round 1

Reviewer 1 Report

The study is devoted to the current and important topic of Performance improvement of predictive model of demolition waste generation rate.

The article is well written, good designed and the problem that the authors are investigating is relevant.

The structure of the article meets the classical requirements.

The article title is clear.

Keywords are chosen correctly and match the direction of the research.

Below are some comments to improve the article.

Abstract. There is no justification for the feasibility of using ANN for forecasting demolition waste. There is also no description of the scientific novelty.

Part of the title is "predictive model" but there is no forecasting. It seems to me that it would be logical to end the article using the proposed method to prediction of demolition waste generation rate on new input data that the network has not yet "seen"

The authors need to justify the choice of all input variables. For example, what effect does Location and Usage have on DWGR.

Why do the authors evaluate the performance of models using pairs of similar metrics? For example, R/R2 and RMSE/MAE. What additional data does this use provide? Please explain.

Formulas 7, 8, 9 are repeated.

Figure 4, 5, 6. The authors need to specify the names for figures a b c

Author Response

Thank you for your review of the completeness of this paper. The revision of this paper reflecting your review is as follows:

Point 1: Abstract. There is no justification for the feasibility of using ANN for forecasting demolition waste. There is also no description of the scientific novelty.

Response 1: Thank you for your review We have revised and supplemented the abstract by reflecting your opinion. Please see the revised abstract.

Point 2: Part of the title is "predictive model" but there is no forecasting. It seems to me that it would be logical to end the article using the proposed method to prediction of demolition waste generation rate on new input data that the network has not yet "seen".

Response 2: Thanks for your comments. For this part, it seems appropriate to use "machine learning model" rather than "predictive model". Therefore, we reflected this in the title.
However, there is one thing I should point out about your opinion. Regarding your opinion about predictive models, it is difficult to agree with the opinion that a predictive model cannot be made without new input data. Because, if your opinion is correct, it seems that it cannot be a predictive model of many existing papers mentioned in this paper. I'd love to hear what you think about this.

Point 3: The authors need to justify the choice of all input variables. For example, what effect does Location and Usage have on DWGR.

Response 3: Thank you for your review In reflection of your opinion, the information on input variable selection has been supplemented. See lines 147 to 153 in section 2.1 data source.

Point 4: Why do the authors evaluate the performance of models using pairs of similar metrics? For example, R/R2 and RMSE/MAE. What additional data does this use provide? Please explain.

Response 4: As you said, it is a similar metric evaluation index, but we sometimes evaluate the performance of the model additionally through the R value when the value of R2 is the same or similar in the performance evaluation of the model. In addition, RMSE/MAE similarly evaluates the error of the model, so if it is difficult to judge from one indicator, it is possible to evaluate which model is better by referring to additional indicators. In this aspect, we showed four model evaluation indicators in this study.

Point 5: Formulas 7, 8, 9 are repeated.

Response 5: Thank you for your review There was a mistake in the submission process. Equations 7, 8, and 9 were modified to suit the contents.

Point 6: Figure 4, 5, 6. The authors need to specify the names for figures a b c

Response 6: Thanks for your point The picture has been modified to reflect your opinion. Please refer to Figures 4, 5, 6, and 7.

Reviewer 2 Report

Dear authors and editorial team,

1. As the last note in your abstract, please provide a "take-home" message. Also, the abstract must respect the journal template and not exceed 200 words.

2. Rearrange keywords alphabetically.

3. Future research maybe can be included in the conclusions

4. Conclusions would be good to be rephrase and extended.

5. The whole article looks more like a report than research.

6. Some images are not exactly of the desired quality, please improve them.

7.  Maybe a summary of some paragraphs would be good, the article is hard to follow.

Regards,

Author Response

Thank you for your review of the completeness of this paper. The revision of this paper reflecting your review is as follows:

Point 1: As the last note in your abstract, please provide a "take-home" message. Also, the abstract must respect the journal template and not exceed 200 words.

Response 1: Thank you for your review We have revised and supplemented the abstract by reflecting your opinion. Please see the revised abstract.

Point 2: Rearrange keywords alphabetically.

Response 2: Thank you for your meticulous review. Based on your comments, the keywords have been arranged in alphabetical order. And keywords have been modified.

Point 3: Future research maybe can be included in the conclusions

Response 3: Thank you for your advice for the completeness of the thesis. We agree with your opinion, and by reflecting your opinion, we supplemented the contents of additional future research in the last paragraph of the conclusion. Please see the last paragraph of the conclusion.

Point 4: Conclusions would be good to be rephrase and extended.

Response 4: Thank you very much for your review. In reflection of your opinion, the conclusion has been revised and supplemented as a whole. Please see revised conclusions in their entirety.

Point 5: The whole article looks more like a report than research.

Response 5: Thank you for your review I agree with you. Therefore, we tried to concisely compress the contents without damaging the overall context in many parts. I reflected your opinion, and setion 2.3.1. Autoencoder, section 4. Discussion and recommendations, and section 5. Conclusions have modified many parts. Please refer to modified section 2.3.1. Autoencoder, section 4. Discussion and recommendations, and section 5. Conclusions.

Point 6: Some images are not exactly of the desired quality, please improve them

Response 6: Thanks for your point The picture has been modified to reflect your opinion. Please refer to Figures 4, 5, 6, and 7.

Point 7: Maybe a summary of some paragraphs would be good, the article is hard to follow.

Response 7: Thank you for your review We have modified and supplemented the sentences in many areas to make it easier for readers to understand. In particular, setion 2.3.1. In Autoencoder, section 4. Discussion and recommendations, and section 5. Conclusions, it was judged that the description was excessively long, and we modified many parts in these parts. As in Response 5, Please refer to modified section 2.3.1. Autoencoders, section 4. Discussion and recommendations, and section 5. Conclusions.

Round 2

Reviewer 1 Report

I thank the authors for the changes made to the article.

My comment on question 2.

I agree with you, most articles that have "model for prediction" in the title do not include a demonstration of the model's performance on new data. As a rule, most studies end with a comparison of metrics (losses, correlation, etc.)

The question arises, why create a model and not check how it works on new data?

Reviewer 2 Report

Dear authors and editorial team,

The paper could be published in this form.